# A Transport Policy Whose Injury Impacts May Go Unnoticed: More Walking, Cycling and Use of Public Transport

**DOI:** 10.3390/ijerph16193668

**Published:** 2019-09-29

**Authors:** Rune Elvik

**Affiliations:** Institute of Transport Economics, Gaustadalleen 21, 0349 Oslo, Norway; re@toi.no

**Keywords:** injury, reporting, modal split, transport policy

## Abstract

It is an objective of transport policy in many countries and cities to promote walking, cycling and the use of public transport. This policy seeks to improve public health and reduce emissions contributing to global warming. It is, however, very likely that more walking, cycling and use of public transport will be associated with an increase in traffic injury. Moreover, it is likely that most of this increase will go unnoticed and not be recorded in official road accident statistics. Official statistics on traffic injury are known to be very incomplete as far as injuries to pedestrians, cyclists and public transport passengers are concerned. This incompleteness is a problem when assessing health impacts of more walking, cycling and travel by public transport. In this paper, studies made in the city of Oslo, Norway (population 700,000) are used to develop numerical examples showing how the estimated real and recorded number of injuries may change when 10% of person km of travel performed by car are transferred to walking, cycling or public transport. It is shown that not more than about 2% of the estimated change in the actual number of injured road users will be recorded by official statistics on traffic injury.

## 1. Background and Research Problem

It is an objective of transport policy in many cities and countries to encourage more walking, cycling and travel by public transport. These changes in travel habits are intended to improve public health and reduce global warming. Walking and cycling are associated with a higher risk of injury per kilometre travelled than travel by car (as driver or passenger). Travel by public transport is safer than travel by car, at least as far as the risk of injury in traffic crashes is concerned. However, public transport rarely takes you door-to-door and involves “last mile” trips made on foot or by bike. The access and egress parts of a door-to-door journey by public transport may be associated with a considerably higher risk of injury than the risk sustained while on board a public transport vehicle [1,2].

There is great uncertainty about how a transfer of trips made by car to walking, cycling or public transport may influence the number of traffic injuries. The principal sources of uncertainty include:The reporting of injuries sustained by pedestrians and cyclists in official statistics on traffic injury is very incomplete. Pedestrian falls are not reportable and are not at all included in official injury statistics.Most injuries sustained by public transport passengers are not sustained in reportable crashes, but in non-collision events, such as falls inside the vehicle or falls when boarding or alighting the vehicle [3]. Very few of these events are reported in official statistics on traffic injury.The risk of injury run by pedestrians and cyclists is not independent of the number of pedestrians or cyclists, but displays a non-linearity often referred to as “safety-in-numbers” [4]. Safety-in-numbers means that the risk to each pedestrian or cycling is reduced the more pedestrians and cyclists there are in traffic.The risk of injury within each group of road users varies substantially according to e.g., age, gender and experience. Changes in the number of injuries will therefore depend on who takes up more walking, cycling or use of public transport.The risk of injury varies between different types of public transport and the traffic environment served by public transport. Changes in the number of injuries will therefore vary, depending on which type of public transport increases and the traffic environment it serves.The accessibility of public transport varies considerably. In city centres, there is usually only a short walk to the nearest public transport stop. In suburbs and the countryside, the distance to the nearest stop may be much longer and the contribution to overall risk from the access and egress parts of a trip can be greater than in a city centre.

To get as close to correct results as possible, a health impact assessment of more walking, cycling and travel by public transport should address all these points, which makes it a large and complex task [5]. One may avoid the problem of incomplete injury reporting by assessing mortality impacts only, but given the large number of unreported injuries, it is a legitimate ambition for a health impact assessment to try to include them. Thus, the main question this paper deals with is:

To what extent will changes in the number of injuries as a result of transfer of trips from car to walking, cycling or public transport be recorded in official statistics on traffic injury?

To answer this question, estimates of the risk of injury based on officially recorded injuries will first be presented. Then, studies of incomplete recording of injuries will be reviewed in order to develop estimates of the real number of injuries. Estimates of the risk of injury based on the recorded numbers and the estimated real numbers will be applied to estimate potential changes in the number of injured road users associated with transferring 10% of person km of travel performed by car in the city of Oslo to walking, cycling and public transport. Finally, estimated changes in the number of injured road users relying on the two sources of data will be compared, in order to assess the extent to which the changes will be reflected in official statistics on traffic injury.

Previous studies of the effects on traffic injury of changes in the modal split of travel have relied on a single source of injury data and not compared police records with other sources of data. Results are mixed, but some studies have found that increased cycling is not associated with an increased number of injuries [6,7].

## 2. Estimates of Injury Risks Based on Official Statistics

Figure 1 shows estimates of injury risk for different modes of transport in Norway, based on different sources of data. For car occupants, cyclists and pedestrians, estimates are based on the national household travel survey 2013–2014 [8] and official traffic injury statistics. The risk to bus passengers refers to 2015–2017 and is based on official injury statistics and estimates of bus travel provided by bus companies. The risk to train passengers refers to 2006–2016 and is based on injury recording and passenger statistics kept by train operators [9].

The risk of injury to train passengers is extremely low. Only 1 fatal and 7 serious injuries were recorded during the 11 years the statistics include; less than one injury per year. It should be noted that slight injuries are not recorded. In road traffic crashes, the slightly injured make up almost 90% of the total. If the distribution of injuries by severity is the same among train passengers as in road traffic crashes, the risk estimate given in Figure 1 should be multiplied by a factor of close to 10. In the numerical examples developed later in the paper, the risk of injury to train passengers is set to 0.002 per million passenger km, rather than the risk of 0.0002 given in Figure 1.

It is seen that walking or cycling involves the highest risk of injury. The risk of injury when cycling is roughly twice the risk of injury when walking.

## 3. Studies of Reporting of Injuries in Official Statistics

### 3.1. Cyclist Injuries

The emergency injury clinic in Oslo (Oslo legevakt), which is part of the university hospital, performed a special recording of injuries to cyclists in 2014 [10]. The clinic treats slight injuries not requiring admission to hospital. Injury victims travel to the clinic on their own (or with help from family or friends) and are normally treated and return home the same day.

In 2014, the clinic recorded 2184 injured cyclists. This included 46 cyclists who were transported directly to hospital and did not visit the clinic. 1673 cyclists were injured in traffic; the rest were injured outside traffic, e.g., when cycling in the forest. The police recorded 125 injured cyclists in Oslo in 2014. The main differences between the injury records of the emergency clinic and the police are that: (1) Injury descriptions are far more detailed in the medical records than in police records; (2) The police geocode injuries; medical records do not. Table 1 shows the reporting rate according to injury severity and type of crash.

The level of reporting is low, both for crashes involving a motor vehicle and for single bicycle crashes. The latter are almost never reported. The overall level of reporting in 2014 was 125/1673 = 7.5%. The number of injured cyclists reported by the police in Table 1 sums to 124, but there was 1 cyclist for which police data did not state injury severity.

### 3.2. Pedestrians

During 2016, the Oslo emergency injury clinic made a special study of pedestrian injury, modelled on the one for cyclists in 2014 [12]. A total of 6309 injured pedestrians were recorded. 6109 were injured when falling; 200 were injured in traffic crashes. Falls are not defined as reportable crashes and are not included in official statistics.

The police recorded 106 injured pedestrians in traffic crashes in Oslo in 2016. The overall level of reporting was 53% (106/200), which is considerably higher than the reporting of cyclist injuries.

4804 of the 6109 pedestrian falls occurred in traffic; the remainder occurred at other locations.

### 3.3. Public Transport Passengers

A survey of injuries to passengers in buses and trams in Oslo was made in 1997 [13]. The study included both traffic crashes and injuries sustained in non-collision events. The non-collision events associated with injuries included sudden braking or acceleration, and sharp turns and falls while boarding or alighting.

During the period 1989–1995, the police recorded 225 injury crashes involving trams. The tram operator recorded 477 injury crashes during the same period. Police reporting was thus 47% (225/477). Police reporting was low for the non-collision injuries. The police recorded 24 such injuries, the tram operator 299, making for a reporting level of 8%.

A synthesis of 11 studies made in different countries [3] shows that the risk to public transport passengers of getting injured in a non-collision event is high. Estimates of risk vary substantially between studies. Table 2 summarises estimates of risk in the 11 studies that were surveyed. The study by Sagberg and Sætermo [13] included both bus and tram. The other studies refer to buses only.

Estimates of risk vary considerably. The risk of falling on board varies from 0.034 to 1.414 per million person km of travel. Summary estimates of the risk vary between 0.283 and 0.529 per million person km. The risk of falling while boarding or alighting varies between 0.037 and 4.495 per million passengers. Summary estimates of risk vary between 0.941 and 1.734 per million passengers.

Several of the studies are old and probably overestimate current risk. In general, the risk of traffic injury has declined rapidly in recent years. This decline in risk most probably also applies to public transport passengers. Two British studies that relied on the same source of data [14,15] (police reports; most likely affected by considerable under-reporting) indicate a decline in risk. The first study [14] relied on data for 1999–2001. The second study [15] relied on data for 2008–2012. The first study estimated the risk of falls on board per million passenger km to 0.202; the second study gave a risk of 0.097. The risk of falling when boarding or alighting was estimated to 0.312 per million passengers in the first study and 0.130 per million passengers in the second study. Thus, risk declined by more than 50% in a period of about ten years.

The Norwegian study [11] was based on data for 1989–1996 for trams and 1981–1993 for buses. Elvik [16] estimated the risk of injury in traffic crashes for different modes of transport for 1980–1993. For buses, injury risk during 1981–1993 was 0.041 injured passengers in traffic crashes per million passenger km. As shown in Figure 1, the risk of injury to bus passengers in traffic crashes during 2015–2017 was 0.012 per million passenger km. Risk has declined considerably.

If it is assumed that the risk of falls on board has declined at the same rate, and current risk in the city of Oslo can be estimated as (0.012/0.041) × 0.667 = 0.195. The risk of falling when boarding or alighting is stated per passenger, not per passenger kilometre. To estimate this risk, mean trip length must be known. As an example, a bus producing 1 million passenger km per year, will have 200,000 passengers per year if each passenger makes a trip of 5 km.

Sagberg and Sætermo [13] assumed a mean trip length of 3.7 km for bus passengers. Their study only included city buses. Ruter, the company that administers public transport in the counties of Oslo (mostly urban) and Akershus (mostly suburban or rural) in Norway states that the current mean trip length of bus passengers is 8.2 km. If Sagberg and Sætermo had assumed a trip length of 5 km rather than 3.7, their estimate of risk would have changed from 0.819 to 1.106 per million passengers. In the numerical examples developed later in the paper, it has been assumed that the current risk of falling when boarding or alighting is 0.30 per million passengers.

The risk of injury in traffic crashes to tram passengers was 0.109 injured passengers per million passenger km during 1989–1993 according to Elvik [16]. This injury risk was higher than the risk to bus passengers during the same period. During 2006–2017, the risk of injury to tram passengers in traffic crashes had been reduced to 0.010, a decline of more than 90% from 1989–1993 [7]. It will be assumed that the risk of injury in non-collision events has been reduced at the same rate. Thus, current risk of injury when falling on board is 0.033 per million passenger km. Current risk of falling when boarding or alighting is 0.135 per million passengers. A current mean trip length of 3.2 km has been assumed [17]. Sagberg and Sætermo [13] assumed a mean trip length of 2.6 km.

No estimates of the risk of non-collision injury for metro or train have been found. The risk of serious injury per million passenger km travelled by metro (data for 2012–2017) has been estimated to the low value of 0.001 [17,18]. All these injuries are stated to have occurred while boarding or alighting. If it is assumed that there are about 10 slight injuries for each serious injury (the ratio in road traffic), injury risk while travelling by metro can be estimated to 0.010 per million passenger km. This represents the total risk associated with non-collision events. The metro in Oslo had no traffic crashes (derailments or collisions) involving personal injury during 2012–2017.

Statistics published by the municipality of Oslo states that there were 8.9 million trips by train in Oslo in 2017. Mean trip length is not stated. However, based on the distances between Oslo Central Station and the most distant stations located inside the city limits of Oslo, a rough estimate of mean trip length of 6.6 km is proposed. It can then be estimated that a total of 59 million passenger km was performed by train in Oslo in 2017. The risk estimate for train travel in Figure 1 refers to serious injuries when boarding or alighting. There were no injuries to train passengers in traffic crashes (derailments or collisions) during 2006–2016. As mentioned before, the adjusted estimate of risk is 0.002. This represents the total risk of injury associated with non-collision events when travelling by train.

## 4. Adjusted Estimates of Risk for Oslo—2015

Based on a review of studies in Section 3, and the updates of these studies, Table 3 presents estimates of the risk of traffic injury in Oslo, based on reported numbers and estimated real numbers.

The differences between the reported and estimated real number of injuries are dramatic for pedestrians, cyclists and public transport passengers. For car occupants, roughly half of injuries are reported. The risk of injury when travelling by bus is higher than when travelling by car. The other public transport modes have a lower risk of injury than cars.

Estimates of risk are not based on data for exactly the same years for all modes of transport, but may be considered as roughly representative of the situation in 2015.

## 5. Assumptions about Safety-In-Numbers

The estimates of risk presented in Table 3 apply, roughly speaking, to the city of Oslo in 2015. Although the data do not refer to the same year for all modes of transport, risk changes little from year-to-year and the main focus of this paper is the effects of changes in the modal split of travel – in particular less car driving and more walking, cycling and use of public transport. Readers are reminded that the estimated real numbers of injuries for pedestrians, cyclists and public transport passengers are all based on studies specifically designed to estimate these numbers. The only numbers that are routinely collected are those collected by the police.

In discussions about the potential impacts on traffic injury of more walking or cycling, safety-in-numbers is often mentioned. Safety-in-numbers refers to a tendency for the risk of injury to each pedestrian or cyclist to decline the more pedestrians or cyclists there are in a traffic system. Elvik and Goel [4] recently reported a systematic review and meta-analysis of studies of safety-in-numbers. These studies consistently indicate safety-in-numbers, although the mechanisms producing safety-in-numbers remain too poorly known to conclude that the relationship is causal. Despite this, it has been assumed that there will be safety-in-numbers if more people walk or cycle. The change in risk to each pedestrian or cyclist as a function of their number has been modelled by means of a risk curve with an elasticity of −0.6 [4]. Figure 2 shows this risk curve.

The initial values of traffic volume and risk have been set to 100. Safety-in-numbers is assumed to apply to pedestrians, cyclists and car occupants. No safety-in-numbers effect has been assumed for public transport passengers, as there are no studies showing non-linearity in risk for public transport passengers as a function of the number of passengers.

## 6. Changes in the Number of Injuries Associated with More Walking, Cycling and Travel by Public Transport

Four scenarios involving more walking, cycling and travel by public transport have been defined. In all of them, travel by car, i.e., person km performed by car, is reduced by 10%, from 5423 million person km per year to 4881 million person km per year. The displaced car trips are replaced by more walking, cycling and travel by public transport. The scenarios are intended to show changes in modal split that are realistic, i.e., within ranges that have been observed historically. They differ principally with respect to how the displaced journeys by car are distributed between walking or cycling and use of public transport. The following four scenarios are considered:The displaced car trips are allocated proportionally to other modes of travel. This means that walking, cycling and public transport all increase by 28.6%. The risk to pedestrians and cyclists then declines by 14%.Displaced car trips are replaced by 20% increase in walking and cycling, and 31.6% increase in public transport. All modes of public transport increase by the same percentage. Risk to pedestrians and cyclist decrease by 10.4%.Displaced car trips are replaced by 10% increase in walking and cycling and 35.1% increase in public transport. All modes of public transport increase by the same percentage. Risk to pedestrians and cyclists decrease by 5.6%.Displaced car trips are replaced by 35% increase in walking and cycling and 26.3% increase in public transport. All modes of public transport increase by the same percentage. Risk to pedestrians and cyclists decrease by 16.5%.

The results are presented in Table 4 and Figure 3. The first row of Table 4 shows the current number of injuries. The following rows show the estimated number of injuries for each of the four scenarios described above. It is seen that changes in the recorded number of injuries are small. The largest change is an increase of 2.9%, associated with 35% more walking and cycling and 26.3% more public transport. In this scenario, the estimated real number of injuries increases by 11.5%—considerably more than the recorded number of injuries. In one scenario, there is a small decline in the recorded number of injuries, but still an increase in the estimated real number of injuries. This is when walking and cycling increases by 10% and public transport by 35.1%.

Figure 3 shows how small the changes in the recorded number of injuries are compared to changes in the estimated real number of injuries. None of the estimated changes in the recorded number of injuries are statistically significant at conventional levels. These small changes may then be interpreted as random variation only, and not as signs of a real increase in the number of injured road users. The estimated increases in the estimated real number of injuries are, however, in all cases highly statistically significant and could thus not be dismissed as merely random variation.

Unless dedicated projects are set up to record injuries sustained by pedestrians, cyclists and public transport passengers, or a greatly improved system of injury recording is established, the only source of data that will be available to monitor the impacts on traffic injury of more walking, cycling and travel by public transport will be the police reports. The scenarios developed indicate that the police will hardly detect any changes in the number of injured road users. Police data will therefore give a very misleading impression of the effects on traffic injury of more walking, cycling and travel by public transport.

## 7. Discussion

It has been known for a long time that official accident statistics are incomplete [20]. The reporting of cyclist injuries is very low. This has been found in studies performed in many countries. There are also a number of studies, the earliest dating from the 1980s, showing that falls among pedestrians is a significant problem [21]. These falls are not defined as reportable traffic crashes. It is not uncommon that the number of pedestrian falls is larger than the number of single vehicle crashes among cyclists.

It is highly improbable that injuries sustained by pedestrians or cyclists in events not involving any other road users will ever be reported very well by the police. Making pedestrian falls a reportable traffic crash will hardly make a difference.

Health care facilities are best placed to record injuries sustained by pedestrians and cyclists. Injury recording should include pedestrian falls, although these are not defined as a road accident. A limitation of health care records, at least in Norway, is that geocoding is missing. By identifying the location where the injury was sustained, the value of the records would increase for users like highway agencies, who would then be able to identify the worst locations and treat them. It ought to be feasible to geocode most injuries. Most of those who are injured are slightly injured and should be able to point out where the injury occurred on a map. The wide availability of digital maps would make it possible to enter geocoding into the same file as other data regarding the injury.

Better recording of injuries sustained by public transport passengers is also needed in order to get a more truthful picture of the injury impacts of more travel by public transport. Many public transport vehicles have surveillance cameras. These are intended to detect vandalism, theft or other unwanted events. If suitably positioned, the cameras would also be able to record falls on board or when boarding or alighting. Routines would need to be established to regularly store and analyse data recorded by the cameras.

In many ways, road safety policy has become more evidence-based in the last 15–20 years than it was earlier. However, the evidence base for a policy designed to increase walking, cycling and travel by public transport is less developed, at least as far as traffic injury is concerned. Clearly, the main objective of the policy is not to improve road safety. It is nevertheless relevant to know the impacts on road safety, in particular because the possibility of an increase in traffic injury is still used as an argument against the promotion of active transport. There is little doubt that the public health benefits or more active transport greatly exceed the costs in terms of traffic injury (see e.g., [22,23]). Yet, it is an undeniable fact that many car users would expose themselves to a higher risk of injury if they switched to walking, cycling or public transport.

## 8. Conclusions

To be unbiased, a health impact assessment of more walking, cycling and travel by public transport should be based on as complete data on injuries as possible. No single source of data is known to be complete, but official statistics on traffic injury is known to be very incomplete. Health care facilities are likely to provide the most complete recording of injuries and the data collected by health care facilities should therefore always be used when performing health impact assessments. Injury severity and long-term impacts of injuries needs to be better known.

A policy recommendation that follows easily from the findings in this paper is that any policy designed to increase walking, cycling or travel by public transport should include measures to make these modes of travel safer. Improving the safety of walking, cycling or travel by public transport is likely by itself to make these modes of travel more attractive.

## Figures and Tables

**Figure 1 ijerph-16-03668-f001:**
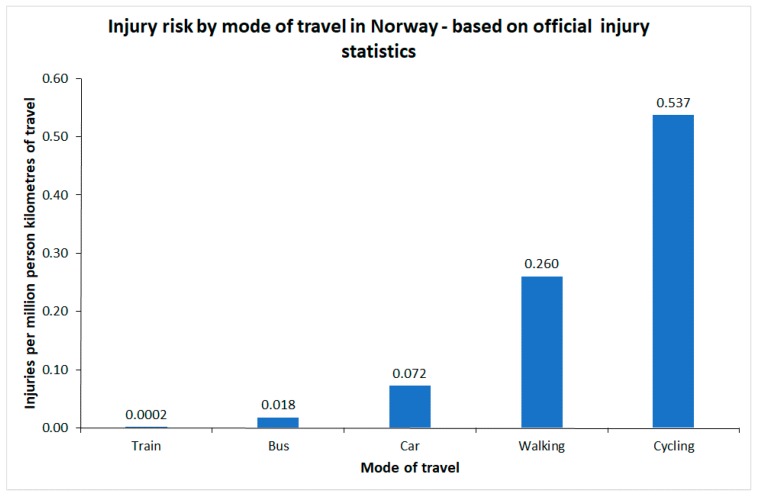
Injury risk by mode of travel in Norway, based on official injury statistics.

**Figure 2 ijerph-16-03668-f002:**
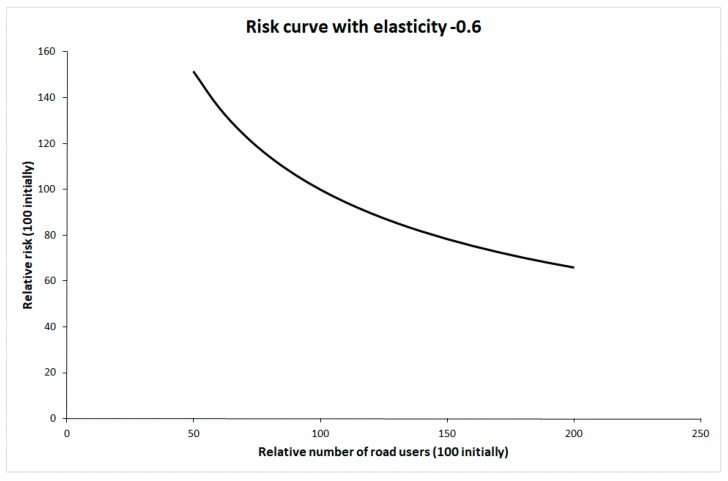
Risk curve with elasticity −0.6.

**Figure 3 ijerph-16-03668-f003:**
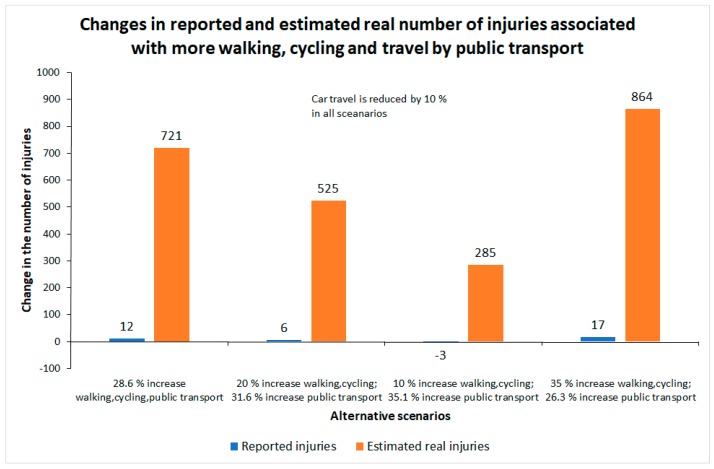
Impacts on traffic injury of four scenarios involving more walking, cycling and travel by public transport.

**Table 1 ijerph-16-03668-t001:** Reporting of cyclist injuries in police statistics by injury severity and type of crash. Oslo 2014. Based on Elvik [11].

Injury Severity (1)	Crash Involving Bicycle and Motor Vehicle			Single Bicycle Crashes		
	Police	Clinic	Reporting (%)	Police	Clinic	Reporting (%)
Slight	100	465	21.5	5	1142	0.4
Serious	18	20	90.0	1	43	2.3
Total	118	485	24.3	6	1185	0.5

(1) In medical data: slight = AIS 1–2 serious = AIS 3+. In police data: slight and serious are defined in terms of specific types of injury.

**Table 2 ijerph-16-03668-t002:** Risk of injury to public transport passengers in non-collision events. Source: Elvik [3].

Study	Country	Falls on Board Per Million Person Km	Falls when Alighting/Boarding Per Million Passengers
Brooks et al. 1980	Great Britain	0.064	0.037
Vaa 1993	Norway	0.036	0.456
Fruin et al. 1994	USA	0.614	1.824
King 1996	USA	0.393	1.502
Sagberg and Sætermo 1997 (tram)	Norway	0.365	1.144
Sagberg and Sætermo 1997 (bus)	Norway	0.667	0.819
Skjøth-Rasmussen et al. 1999	Denmark	0.729	1.063
Kirk et al. 2003	Great Britain	0.202	0.312
Bjørnstig et al. 2005	Sweden	0.218	1.927
Halpern et al. 2005	Israel	0.159	0.497
Strathman et al. 2010	USA	1.414	4.495
Barnes et al. 2016	Great Britain	0.097	0.130
Summary estimates			
Simple mean	All	0.413	1.184
Weighted mean	All	0.283	1.091
Median	All	0.292	0.941
Studies with good data	Six of twelve	0.529	1.734

**Table 3 ijerph-16-03668-t003:** Police reported and estimated true number of traffic injuries in the city of Oslo, Norway, during one year (representative for about 2015).

Mode of Transport	Reported Number of Injuries	Estimated Real Number of Injuries	Million Person Km	Reported Injuries Per Million Person Km	Real Injuries Per Million Person Km
Walking	106	5004	333	0.318	15.027
Cycling	125	1673	158	0.791	10.589
Bus (1)	6	129	477	0.012	0.270
Tram (2)	2	14	165	0.010	0.085
Metro (3)	0	7	706	0.000	0.010
Train (4)	0	0	59	0.000	0.002
Car (5)	348	660	5423	0.064	0.122

(1) The reported number of injuries is estimated by relying on the risk estimates of 0.012 per million person km. The estimated real number includes non-collision injuries in addition to injuries in traffic crashes. (2) The reported number of injuries is estimated by relying on the risk estimates of 0.010 per million person km. The estimated real number includes non-collision injuries in addition to injuries in traffic crashes. (3) All injuries are assumed to occur in non-collision events. Based on statistics by Ruter. (4) All injuries are assumed to occur in non-collision events. (5) The reported number of injuries is based on police statistics for 2014. Person km was estimated by multiplying the number of cars in Oslo (257,014) by mean annual driving distance (12,411 km) and mean car occupancy rate (1.7) [19].

**Table 4 ijerph-16-03668-t004:** Estimated changes in the number of injured road users in Oslo associated with more walking, cycling or travel by public transport.

Change in Car Travel	Change in Walking/Cycling	Change in Public Transport	Recorded Number of Injuries	Percent Change	Estimated Real Number of Injuries	Percent Change
None	None	None	587		7487	
−10%	+28.6%	+28.6%	599	+2.0%	8208	+9.6%
−10%	+20%	+31.6%	593	+1.0%	8012	+7.0%
−10%	+10%	+35.1%	584	−0.5%	7772	+3.8%
−10%	+35%	+26.3%	604	+2.9%	8351	+11.5%

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
