# Peer review of "A Transport Policy Whose Injury Impacts May Go Unnoticed: More Walking, Cycling and Use of Public Transport"

_ijerph, 2019, doi:10.3390/ijerph16193668_

Round 1
Reviewer 1 Report
Interesting topic which fills an important gap in methodology of transport policy process, sound interpretation, clear presentation of results. Author touched the spot of relevance for injury severity, where further researches on this could contribute to raise scientific soundness of the paper and to have more precise conclusion and policy implications. Somehow, this should be mentioned in the paper. Anyway, this shortcoming doesn't depreciate the main findings.
Author Response
I have added a note on the need for more research on injury severity in the paper.
Reviewer 2 Report
The paper clearly deals with the incompleteness of database on traffic injures especially for pedestrians, cyclists and public transport passengers
Broadly speaking,
(3.1)I would better specify the characteristics of hospital vs police databases; Table 1 - I would describe "slight"vs "serious" (in a footnote?) 6 - I would specify how you "built" the 4 scenarios I would also add in the conclusions: a clear summary of the paper issues (modal shift---> more/less risks ---> importance of data availability for policy) + some policy suggestions in terms of road safety (for all).Minor problems:
(3.2) pedestrians falling at home are still pedestrians? Table 3 - source of note 3? check in note 5 "ION"instead of "on".
Author Response
I am not sure what "better specify characteristics of hospital vs police databases" means. I have, however, added the following sentence to the paper:
The main differences between the injury records of the emergency clinic and the police are that: (1) Injury descriptions are far more detailed in the medical records than in police records; (2) The police geocode injuries; medical records do not.
I have added a footnote on the definition of slight and serious injuries in Table 1.
I have added the following note on how the scenarios were developed:
The scenarios are intended to show changes in modal split that are realistic, i.e. within ranges that have been observed historically. They differ principally with respect to how the displaced journeys by car are distributed between walking or cycling and use of public transport.
In the conclusions, I have added the following:
A policy recommendation that follows easily from the findings in this paper is that any policy designed to increase walking, cycling or travel by public transport should include measures to make these modes of travel safer. Improving the safety of walking, cycling or travel by public transport is likely by itself to make these modes of travel more attractive.
I have deleted the reference to "home" where pedestrians fall.
The source of note 3 in Table 3 is statistics kept by Ruter
The error in note 5 has been corrected.
Reviewer 3 Report
The author raises an very important issue related to the hidden effects of developing environmentally friendly forms of travel. I believe that the article is part of the most important transport issues undertaken by researchers today. In my opinion, the article is well prepared although it is mainly descriptive. I think that the structure is also well prepared. Among the issues that, in my opinion, require some supplements, is the issue of reference to other transport researches in this subject. The article somewhat lacks a literature review of similar studies. I urge the author to broaden the review of literature, after all, he is an expert in this field.
I rate the article positively and I think it is suitable for publication in a journal.Author Response
I thank the reviewer for the positive comments on the paper. I am not aware of studies that are similar to this one, but I have added a short section quoting some studies of impacts on traffic injury of increased cycling and increased use of public transport.